# Kinematics Analysis and Gait Study of Bionic Turtle Crawling Mechanism

**DOI:** 10.3390/biomimetics9030147

**Published:** 2024-02-28

**Authors:** Zhuo Wang, Wanlang Peng, Bo Zhang

**Affiliations:** College of Mechanical and Electrical Engineering, Harbin Engineering University, Harbin 150001, China; wangzhuo_heu@hrbeu.edu.cn (Z.W.); zhangbo_heu@hrbeu.edu.cn (B.Z.)

**Keywords:** drainage culvert desilting, underwater dredging crawler mechanism, foot-end trajectory planning, gait optimization, energy consumption characteristics

## Abstract

Longer distance water delivery culverts pose obstacles such as deposited silt, stones, and dead trees. In this paper, a crawling robot is designed to mimic the joint structure of a turtle using bionic design principles. The mechanism and gait of the robot are analyzed. The kinematics model of the robot is established using the D-H method and analytical approach, while the dynamics model is established using Lagrange’s method. Based on kinematics and dynamics analysis theory, compound cycloid and cubic polynomial motion trajectories for the robot foot are planned along with a crawling gait resembling that of a turtle’s abdomen. Simulation experiments and scale prototype experiments confirm that when gait parameters are identical, the energy consumption of compound cycloid trajectory exceeds that of cubic polynomial foot trajectory. When planning these two types of foot trajectories, it was observed that energy consumption ratio decreases with increasing step length but increases with increasing step height.

## 1. Introduction

In the South-to-North water diversion project in China, there are many long-distance water transfer culverts with complicated internal environment, such as deposited silt, stones, dead trees and other obstacles. In order to ensure the delivery of higher quality water quality, at present it is necessary to manually operate silt and other debris cleaning equipment, which is very inconvenient, and it is in urgent need of a robot that can work in a harsh silt environment. To solve this problem, the South-to-North Water diversion project proposed the following research topic: “Research on key technologies and equipment for emergency rescue and rapid repair of South-to-North Water Diversion Project”.

This study is based on the research on desilting schemes of water transport buildings and flood discharge buildings in the Research on Key Technologies and Equipment for Emergency Rescue and quick Repair of South-to-North Water Transfer Project, which is the fourth topic of the proposed “Research on Key Technologies and Equipment for Building silting Removal”. According to its moving mode, robots can be divided into wheeled, tracked and legged robots [1]. There are many terrains in nature that cannot be traversed by conventional wheeled or tracked robots, but can be easily traversed by foot robots. As a kind of legged robot, a quadruped robot has more advantages than biped and hexapod robots [2,3,4,5].

By improving the land robot and adding waterproof treatment, we can get a robot that can walk underwater. This paper mainly studies the design and gait problems of the four-legged bionic turtle underwater robot. The existing research on the underwater foot robot focuses on imitating the crabs, lobsters and other multi-legged reptiles, which have a small carrying capacity. It is difficult to install underwater operation equipment, and the control systems involved are complicated.

In 2000, Koray K. A and others at Northeastern University developed a lobster-like robot [6]. In 2004, Tanaka T and others at the Harbor Institute of Technology in Japan developed an amphibious robot [7,8]. In 2005, Akihisa et al. from Yakin University in Japan developed the turtle-like imitation Turtle robot; in 2010, Kato N et al. from Osaka University in Japan developed the RT-I robot robot. In 2010, Naro-Tartaruga imitation turtle robot developed by ETH Zu and others at ETH Zurich, Switzerland [9].

In 2011, Jun B and others at the Korea Institute of Ocean Research developed a CR200 crab-like robot that can walk within 200 m [10,11].

In 2013, Low, K. H et al. from School of Aerospace Engineering and Mechanical Engineering of Nanyang Technological University in Singapore developed MiniTurtle-I imitation turtle robot [12].

In 2017, Kim H and others from South Korea’s Central South National University designed a bionic beetle called CALEB10: d.Bobot robot [13].

In 2021, Grezmak J et al. from Case Western Reserve University in the United States designed a hexapod bionic crab robot [14].

In 2022, Professor Guo Shuxiang from Beijing Institute of Technology and Xing Huiming from Harbin Engineering University designed a bionic turtle amphibious spherical robot with high mobility, high concealability, miniature compactness and multi-mode movement [15,16].

As a crucial mobile carrier of underwater equipment, underwater robots can be equipped with operational machinery, sample collection devices, environmental sensors, and other equipment to accomplish diverse marine scientific research tasks [17,18,19,20]. Based on the current structural characteristics, gait planning, and motion control methods of terrestrial legged robots, amphibious robots, and underwater robots. This paper presents the design of a foot bionic turtle desilting robot suitable for water delivery culverts or culverts. Additionally, it establishes a mechanical system energy consumption model for the bionic turtle desilting robot while explaining energy consumption evaluation indicators and exploring its optimal gait pattern with low energy consumption.

## 2. Overall Scheme Design

A sea turtle can frequently be observed emerging from the water and making its way towards the beach on a summer evening, supporting itself alternately with its feet and belly, while propelling its body by swinging its limbs back and forth, as depicted in Figure 1. In addition to being capable of standing and walking on soft sand, turtles possess remarkable load-bearing characteristics and proficient crawling abilities. Turtles primarily inhabit water due to their limb fins, only venturing ashore during the breeding season. They support themselves alternatively with their feet and abdominal region while moving their body through rhythmic swings of their limbs, as illustrated in Figure 2.

The turtle crawling action is divided into four parts: legs placement, crawling, legs lifting, and legs swinging. Legs placement refers to the downward movement of the limbs to contact the ground; crawling indicates the relative motion between the foot end and the ground; legs lifting refers to the upward movement of the limbs until the abdomen is on the ground; legs swinging refers to the swinging of the limbs in the air. According to the turtle crawling action, the sequence relationship of leg movements during the turtle crawling process can be represented by Figure 3 [21]. Where 0 represents the foot end or body abdomen in the support phase, and 1 represents the foot end or body abdomen in the swing phase. The body is first supported by the body abdomen with 0, and the foot ends of the limbs are in a suspended swinging state with 1. Then, the limbs gradually land, and the body abdomen separates from the ground. Under the friction force between the foot end and the ground, the body slides forward. After sliding, the limbs gradually lift up and the body abdomen gradually lands. When the body abdomen is fully in contact with the ground in the support phase 0, and the foot ends of the limbs are in a suspended swinging phase 1, a complete periodic motion gait is formed. This gait has high anti-tilt characteristics.

As shown in Figure 4, it is a partial bone structure diagram of the turtle. The analysis of the single leg bone structure of the turtle shows that the single leg of the turtle has a spherical pair and two rotational pairs. The simplified motion mechanism model is shown in Figure 5. There are high pairs in the joint structure of the turtle, which also makes its movement more flexible. If the bionic turtle dredging robot also adopts a high pair structure, it will reduce its robot’s load-bearing characteristics, which is not conducive to installing underwater dredging equipment. In order to increase the load-bearing characteristics of the robot, the single leg structure of the bionic turtle dredging robot is designed to be arranged parallel to the direction of travel of the body. The supporting leg mimics the radius and ulna of the turtle, and the swinging leg mimics the humerus of the turtle. Each leg is designed with two joints. The schematic diagram of the leg of the bionic turtle dredging robot is shown in Figure 6, on which the entire structure of the robot can be designed.

The most front end of the robot is a desilting execution device, which is mainly composed of rotary reamer, mud pump and truss. The mud can be crushed by the high-speed rotary reamer, and the mud pump is fixed on the truss to suck out the crushed mud. The turtle crawling and dredging walking mechanism has four walking feet and a rear steering foot. The walking foot has two degrees of freedom on one leg and is powered by a pair of electric cylinders. The rear side steering foot also has two degrees of freedom and is powered by electric cylinders. The robot walks, turns, and reverses through the length expansion and transformation of the leg electric cylinder, as shown in Figure 7.

In addition, the robot’s control chip and visual processing module are installed inside the underwater electronic chamber, and the visual acquisition module can send the real-time working status back to the operator, so as to monitor the working status and adjust the walking gait in time.

## 3. Kinematics and Dynamic Analysis

### 3.1. Forward Solution of Kinematics

Kinematic analysis can provide the basis for foot trajectory planning and motion programming. Based on the D-H parameter transformation, the joint coordinate system was established, and the kinematics model of the robot was simplified to the kinematics model of the leg end plantar and the body center [22], as shown in Figure 8. The four legs of the robot have the same structure. When the robot is walking in a straight line, the kinematic transformation relationship of each leg is the same. Therefore, this paper establishes the kinematic transformation relationship of its single leg.

After establishing the kinematic model of the desilting robot, the corresponding motion pose transformation can be solved. The parameter variables of the robot are shown in Table 1.

By substituting the corresponding data, the coordinates of the robot foot in the geodetic coordinate system can be obtained through the above coordinate transformation relation. The transformation between the central coordinate system ∑*_M_* and the foot coordinate system ∑*_F_* can be obtained by multiplying the transformation matrix described above, and its pose matrix can be represented by Equation (1).
(1)TFM=T0M⋅T10⋅T21⋅TF2=[cos(θ1+θ2)0sin(θ1+θ2)l1cos(θ1+θ2)+s1cosθ1+a−l010−w−b−sin(θ1+θ2)0cos(θ1+θ2)−l1sin(θ1+θ2)−s1sinθ10001]

According to Formula (1), the coordinates (*^M^X_F_*, *^M^Y_F_^M^Z_F_*) of the foot in the robot body coordinate system can be determined. Forward kinematics solution is to solve the workspace at the end of the foot of a single leg when the joint variables are known. Assuming that the swing angle of the supporting leg ranges from −0.75π to −0.25π and the elongation of the supporting leg is 700 mm, the working space during the swing of the supporting leg is drawn with the rotation center of the supporting leg as the origin, as shown in Figure 9.

The drawn area is not a complete sector area, which is determined by the structural characteristics of the robot designed. When the support leg is shortened to the ultimate position, it has a certain distance from the center of rotation, and cannot coincide with the center of rotation. Constants and variables are substituted into Equation (1), and extreme values of parameters in the oscillating process are taken as verification. The obtained foot coordinates satisfy the drawn workspace region. As can be seen from Figure 4, the swing step of the supporting leg in the *x*-axis direction ranges from −0.4 m to 0.4 m, and the lifting height in the *z*-axis direction ranges from 0.3 m to 1 m.

### 3.2. Inverse Kinematics Solution

The inverse kinematics solution can obtain the relationship between the coordinate of foot, swing angle of support leg and the length of electric cylinder. It can provide reference for the selection of electric cylinder and the determination of step size and height in subsequent gait planning. Since the degree of freedom of the single leg of the robot is less, the kinematics can be solved by geometric method directly. The purpose of solving the inverse kinematics in this paper is to study the relationship between *θ*_1_ angle variable and the flex length variable *s*_2_, that is, given the parameter variation range of *θ*_1_ angle variable, the variation range of the flex length variable *s*_2_ can be solved. The leg structure of the robot is the same, and the supporting leg and swing of the robot are simplified. Figure 10 shows the simplified relationship diagram of the geometric single-leg structure of the robot [23].

In Figure 5, the fixed position of the swinging leg and the body is set as point A, the connection between the swinging leg and the supporting leg is set as point B, the projection position of the hinged point of the supporting leg relative to the supporting leg is set as point C, and the fixed position of the supporting leg and the body is set as point O_0_. Let the angle between O_0_C and O_0_B be *y*_1_, and the angle between O_0_A and O_0_*x*_0_ be *y*_2_. Let the coordinate of point B be (*x*_b_, *y*_b_) and the coordinate of point C be (*x_c_*, *y*_c)_. From the geometric relation in Figure 5, it can be obtained: *y*_1_ = arctan(*d*_1_/*d*_2_), *y*_2_ = arctan[*h*/(*l* − *a* − *c*)]. The swing angle of the supporting leg *θ*_1_ can be calculated by Equation (2).
(2)$$\begin{array}{ll} \theta_{1} & =\arccos\frac{{s_{2}}^{2}-{d_{1}}^{2}-{d_{2}}^{2}-{\left(l-a-c\right)}^{2}-h^{2}}{2\cdot \sqrt{\left({d_{1}}^{2}+{d_{2}}^{2})\cdot [{\left(l-a-c\right)}^{2}+h^{2}\right]}}-\gamma_{1}+\gamma_{2}\\ & =\arccos\frac{{s_{2}}^{2}-2.4674}{1.632}-0.0842{}^{\circ} \end{array}$$

According to Equation (2), the curve relationship between variables *θ*_1_ and *s*_2_ can be obtained, as shown in Figure 11.

When the length of the swinging leg *s*_2_ is between 1.1 m and 1.9 m, it can be obtained that the swinging angle of the supporting leg is between −0.7π and −0.25π. The change of the swing length plays a decisive role in the swing angle. In order to meet the needs of the swing angle, the length parameters of the variable part of the electric cylinder selected in this paper should meet the above interval.

After obtaining the relationship between the swing joint angle of the supporting leg and the expansion length of the electric cylinder of the swinging leg *s*_2_, in order to ensure the accuracy of gait planning and prevent the situation that there is no solution for the given foot coordinates and the predetermined trajectory planning cannot be achieved, it is also necessary to determine the relationship between the robot foot coordinates and the corresponding expansion length variables *s*_1_ and *s*_2_ of the electric cylinder. The forward kinematics analysis of the robot can obtain the position coordinates of the robot foot in the central coordinate system of the robot body, as shown in Equation (3).
(3){PXM=l1cos(θ1+θ2)+s1cosθ1+a−lPYM=−w−bPZM=−l1sin(θ1+θ2)−s1sinθ1

In the formula, the relation between the angle of *θ*_2_ and *β*_F_ and *θ*_1_ is *θ*_2_ = 90° + *β_F_* − *θ*_1_; *β_F_* is the dip angle between the foot and the ground.

Since the angle of *θ*_2_ is determined by *β*_F_ and *θ*_1_, the relationship between the variable *s*_1_ of the electric cylinder expansion length and the coordinate parameters of the foot can be obtained by substituting it into the simplified Equation (2), as shown in Equation (4), and the relationship between the angle of the swinging joint of the support leg and the coordinate parameters of the foot is shown in Equation (5).
(4)s1=PXM+l−a−l1cos(θ1+θ2)cosθ1=PXM+l−a+l1sinβFcosθ1
(5)θ1=arcsin−l1sin(θ1+θ2)−PZMs1=arcsinl1cosβF−PZMs1

By solving Equations (2) and (5) simultaneously, the analytical formula of the *x* and *y* coordinates of the foot and the length *s*_1_ of the electric cylinder can be obtained, as shown in Equation (6):(6)s1=(PZM+l1sin(θ1+θ2))2+(PXM+l−a−l1cos(θ1+θ2))2

By solving Equations (2), (4) and (5) simultaneously, the analytical formula of foot coordinates and electric cylinder length *s*_2_ can be obtained, as shown in Equation (7):(7)s2=1.632×cos(−l1sin(θ1+θ2)−PZMs1+0.0842°)+2.4674

The analytical derivation process of the coordinate of the foot of the robot’s front legs and the corresponding expansion length of the electric cylinder *s*_1_ and *s*_2_ is the same as above, but the coordinate symbols are opposite. The trajectory planning of the robot foot involves the *x* coordinate and *z* coordinate of the robot foot. In order to determine the planning range of coordinate parameters, it is necessary to explore the relationship between the coordinate of the foot and the length of the electric cylinder of the robot leg.

According to Equation (6), the relationship between the coordinate of the foot and the supporting leg *s*_1_ can be obtained, as shown in Figure 12a. According to Equation (7), the relationship between the coordinate of the foot and the swinging leg *s*_2_ can be obtained, as shown in Figure 12b.

Figure 12a plots the relationship between the coordinates of the internal foot and the distance *s*_1_ of the two joints in the *X*-axis coordinate interval [−2 m, −1 m] and *Z*-axis coordinate interval [−0.2 m, −1 m]. It can be seen from the figure that the length of the supporting leg *s*_1_ is jointly determined by the *X*-axis and *Z*-axis coordinates, in which the change of *Z*-axis coordinates has a great influence on the supporting leg *s*_1_. The three-dimensional surface is symmetrical at the *X*-axis coordinate of about 1.5 m, which is consistent with the working area obtained from the forward kinematics analysis of the supporting leg. The range of distance between the two joints is about 0.8 m.

Figure 12b draws the relationship between the coordinate of the foot and the swinging leg *s*_2_. According to the robot leg motion model, when the length of the swinging leg remains unchanged, a cluster of corresponding z coordinate and *x* axis coordinate can be obtained, and curves satisfying the conditions can be found in the figure. When the *X*-axis coordinate of the robot foot is located at 1.5 m, the length of the swinging leg is independent of the *Z*-coordinate, which conforms to the three-dimensional structure design of the robot leg. As can be seen from the figure, the length of the robot’s swinging leg *s*_2_ is greatly affected by the *x* coordinate, and the length transformation range of the swinging leg *s*_2_ is from 1.1 m to 1.9 m. When the gait planning foot coordinates are within the *X*-axis coordinate interval [−2 m, −1 m] and *Z*-axis coordinate interval [−0.2 m, −1 m], as long as the selection of electric cylinder elongation conforms to the range of *s*_2_, the robot foot can reach the predetermined trajectory.

### 3.3. Dynamic Analysis

When the body is in contact with the ground and one leg swings forward, the single leg of the robot only bears its own weight, while the sole of the robot is kept parallel to the ground under the action of gravity. In this case, the single leg of the robot has a swing degree of freedom and a variable length of flex degree of freedom. According to the three-dimensional model of the robot, the centroid position of the single leg is evaluated. Half of the length of the supporting leg is about the centroid *m*_1_ of the supporting leg, and 3/4 of the length of *L*_2_ is about the centroid *m*_2_ of the foot. A coordinate system is established at the swing turning point of the supporting leg, and the dynamic mathematical model of the single-leg swing of the robot is shown in Figure 13 [24].

In Figure 8, the centroid of the supporting leg *m*_1_, the centroid of the foot *m*_2_, the fixed length of the supporting leg *L*_1_, the length of the foot *L*_2_, *L*_3_ is the distance between the hinge point of the supporting leg and the swing leg and the origin of the coordinate system, *φ* is the Angle of the supporting leg, Δ*L* is the variable length of the supporting leg.

The Lagrange method is applied to establish a dynamic analysis model for the single leg of a robot. Firstly, Lagrange equation *L* of the single leg system should be constructed, which can be expressed as:(8)$$L(q,\dot{q})=E_{k}(q,\dot{q})-E_{p}(q)$$

In Equation (9), *E_k_* is the kinetic energy of all rods, and *E_p_* is the potential energy of all rods.

If it is necessary to analyze the dynamics of a single leg, the kinetic energy and potential energy of a single leg should be calculated first. As shown in Figure 8, the coordinates of the supporting leg and the centroid of the foot under the established coordinate system can be obtained. The Angle of the supporting leg and the variable length of the supporting leg are expressed by the generalized coordinates *q*_1_ and *q*_2_ respectively, and then the centroid coordinates of the supporting leg can be obtained as shown in Equation (9) and the coordinates of the foot as shown in Equation (10):(9)$$\left\{ \begin{array}{l} {x_{m}}_{1}=\left(\frac{L_{1}+q_{2}}{2}-L_{3}\right)\cdot \sin q_{1}\\ {z_{m}}_{1}=-\left(\frac{L_{1}+q_{2}}{2}-L_{3}\right)\cdot \cos q_{1} \end{array}\right.$$
(10)$$\left\{ \begin{array}{l} {x_{m}}_{2}=\left(L_{1}+q_{2}-L_{3}\right)\cdot \sin q_{1}+\frac{3}{4}L_{2}\cdot \sin\beta \\ {z_{m}}_{2}=-\left(L_{1}+q_{2}-L_{3}\right)\cdot \cos q_{1}-\frac{3}{4}L_{2}\cdot \cos\beta \end{array}\right.$$

After obtaining the above centroid coordinate function relation, the velocity component at the centroid along the coordinate axis can be obtained by differentiating it, as follows:(11)$$\left\{ \begin{array}{l} {\dot{x}}_{m1}=\left(\frac{L_{1}+q_{2}}{2}-L_{3}\right)\cdot \cos q_{1}\cdot {\dot{q}}_{1}+\frac{\sin q_{1}}{2}\cdot {\dot{q}}_{2}\\ {\dot{z}}_{m1}=\left(\frac{L_{1}+q_{2}}{2}-L_{3}\right)\cdot \sin q_{1}\cdot {\dot{q}}_{1}-\frac{\cos q_{1}}{2}\cdot {\dot{q}}_{2} \end{array}\right.$$
(12)$$\left\{ \begin{array}{l} {\dot{x}}_{m2}=\left(L_{1}+q_{2}-L_{3}\right)\cdot \cos q_{1}\cdot {\dot{q}}_{1}+\sin q_{1}\cdot {\dot{q}}_{2}\\ {\dot{z}}_{m2}=\left(L_{1}+q_{2}-L_{3}\right)\cdot \sin q_{1}\cdot {\dot{q}}_{1}-\cos q_{1}\cdot {\dot{q}}_{2} \end{array}\right.$$

According to the kinetic energy formula, the kinetic energy *E_km_*_1_ of the supporting leg and the kinetic energy *E_km_*_2_ of the foot can be calculated, as shown in Equation (13):(13)$$\left\{ \begin{array}{l} {E_{km}}_{1}=\frac{1}{2}m_{1}\left[{\left(\frac{L_{1}+q_{2}}{2}-L_{3}\right)}^{2}{{\dot{q}}_{1}}^{2}+\frac{{{\dot{q}}_{2}}^{2}}{4}\right]\\ {E_{km}}_{2}=\frac{1}{2}m_{2}\left[{\left(L_{1}+q_{2}-L_{3}\right)}^{2}{{\dot{q}}_{1}}^{2}+{{\dot{q}}_{2}}^{2}\right] \end{array}\right.$$

According to the potential energy formula, the potential energy *E_Pm_*_1_ of the supporting leg and *E_Pm_*_2_ of the foot are shown in Equation (14):(14)$$\left\{ \begin{array}{l} {E_{Pm}}_{1}=-m_{1}g\left(\frac{L_{1}+q_{2}}{2}-L_{3}\right)\cdot \cos q_{1}\\ {E_{Pm}}_{2}=-m_{2}g\left[\left(L_{1}+q_{2}-L_{3}\right)\cdot \cos q_{1}-\frac{3}{4}L_{2}\cdot \cos\beta \right] \end{array}\right.$$

The total kinetic energy of a single leg is obtained by adding *Ekm*_1_ and *Ekm*_2_, and the total potential energy of a single leg is obtained by adding *E_Pm_*_1_ and *E_Pm_*_2_. After calculating the kinetic energy and potential energy of a single leg, Lagrangian function *L* can be obtained according to Equation (15), as follows:(15)$$\begin{array}{ll} L= & \frac{1}{2}m_{1}\left[{\left(\frac{L_{1}+q_{2}}{2}-L_{3}\right)}^{2}{{\dot{q}}_{1}}^{2}+\frac{{{\dot{q}}_{2}}^{2}}{4}\right]\\ & +\frac{1}{2}m_{2}\left[{\left(L_{1}+q_{2}-L_{3}\right)}^{2}{{\dot{q}}_{1}}^{2}+{{\dot{q}}_{2}}^{2}\right]\\ & \begin{array}{l} +m_{1}g\left(\frac{L_{1}+q_{2}}{2}-L_{3}\right)\cdot \cos q_{1}\\ +m_{2}g\left[\left(L_{1}+q_{2}-L_{3}\right)\cdot \cos q_{1}-\frac{3}{4}L_{2}\cdot \cos\beta \right] \end{array} \end{array}$$

We take the derivative of the Lagrange function *L* with respect to the generalized coordinates $q_{1}$:(16)$$\frac{\partial L}{\partial q_{1}}=\left[m_{1}g\left(\frac{L_{1}+q_{2}}{2}-L_{3}\right)+m_{2}g\left(L_{1}+q_{2}-L_{3}\right)\right]\cdot \sin q_{1}$$

We take the derivative of the Lagrange function *L* with respect to the generalized coordinates $q_{2}$:(17)$$\frac{\partial L}{\partial q_{2}}=\left[\frac{1}{2}m_{1}\left(\frac{L_{1}+q_{2}}{2}-L_{3}\right)+m_{2}\left(L_{1}+q_{2}-L_{3}\right)\right]{{\dot{q}}_{1}}^{2}+\left(\frac{1}{2}m_{1}g+m_{2}g\right)\cdot \cos q_{1}$$

By differentiating Lagrange function *L* with respect to $q_{1}$ and then with respect to time *t*, we can obtain:(18)$$\begin{array}{l} \frac{d}{dt}\frac{\partial L}{\partial {\dot{q}}_{1}}=\left[m_{1}{\left(\frac{L_{1}+q_{2}}{2}-L_{3}\right)}^{2}+m_{2}{\left(L_{1}+q_{2}-L_{3}\right)}^{2}\right]\cdot {\ddot{q}}_{1}\\ \;+\left[m_{1}\left(\frac{L_{1}+q_{2}}{2}-L_{3}\right)+2m_{2}\left(L_{1}+q_{2}-L_{3}\right)\right]\cdot {\dot{q}}_{1}\cdot {\dot{q}}_{2} \end{array}$$

By differentiating Lagrange function *L* with respect to $q_{2}$ and then with respect to time *t*, we can obtain:(19)$$\frac{d}{dt}\frac{\partial L}{\partial {\dot{q}}_{2}}=\left(\frac{1}{4}m_{1}+m_{2}\right)\cdot {\ddot{q}}_{2}$$

After obtaining the differential equation of the Lagrange function with respect to the generalized coordinates above, the function expression of the generalized force *Q* of the one-legged robot dynamics model can be written out according to Equation (20):

The driving torque *M* required by joint oscillation is:(20)$$\begin{array}{ll} Q_{1}= & \left[m_{1}{\left(\frac{L_{1}+q_{2}}{2}-L_{3}\right)}^{2}+m_{2}{\left(L_{1}+q_{2}-L_{3}\right)}^{2}\right]\cdot {\ddot{q}}_{1}\\ & +\left[m_{1}\left(\frac{L_{1}+q_{2}}{2}-L_{3}\right)+2m_{2}\left(L_{1}+q_{2}-L_{3}\right)\right]\cdot {\dot{q}}_{1}{\dot{q}}_{2}\\ & +\left[m_{1}g\left(\frac{L_{1}+q_{2}}{2}-L_{3}\right)+m_{2}g\left(L_{1}+q_{2}-L_{3}\right)\right]\cdot \sin q_{1} \end{array}$$

*F* is the driving force required for joint oscillation:(21)$$\begin{array}{ll} Q_{2} & =\left(\frac{1}{4}m_{1}+m_{2}\right)\cdot {\ddot{q}}_{2}+\left(\frac{1}{2}m_{1}g+m_{2}g\right)\cdot \cos q_{1}\\ & +\left[\frac{1}{2}m_{1}\left(\frac{L_{1}+q_{2}}{2}-L_{3}\right)+m_{2}\left(L_{1}+q_{2}-L_{3}\right)\right]\cdot {{\dot{q}}_{1}}^{2} \end{array}$$

In order to get the relationship between the single-leg swing driving force and torque over time when the robot body landed on its belly, simplified dynamic analysis was carried out in this part. Set *m*_1_ to 25 kg, *m*_2_ to 5 kg, *L*_1_ to 0.8 m, and *L*_3_ to 0.5 m. Since *Q*_1_ and *Q*_2_ function expressions are nonlinear functions, the input signals of generalized coordinates *q*_1_ and *q*_2_ are simplified as ramp and step functions and brought into the solution module of MATLAB-Simulink to obtain the changes of generalized forces over time, as shown in Figure 14.

The sudden change of driving force and torque is because the given signal is a step signal, and the sudden change of generalized force will be generated instantaneously. The change in direction of the driving force is caused by the leg first lifting off the ground to overcome its own weight and then extending backwards to make contact with the ground. The change of driving torque overcomes the moment of inertia during the swing of the leg and the change of driving torque is due to the change of the direction of the force during the swing.

## 4. Gait Planning and Energy Consumption Characteristics of Robot

### 4.1. Crawling Gait of Imitation Turtle Belly

In order to realize the flexible motion characteristics of robots, existing researchers have found rules from the movement of organisms. A variety of bionic locomotion gaits have been summarized through constant exploration and study of animal locomotion gaits in nature. According to the structural characteristics of the designed robot, the researchers improve the original biological gait and apply the improved bionic gait to the gait planning of the robot [25]. The gait of the robot is designed as the crawling gait of the imitation turtle belly. The leg phase of the robot can be shown in Figure 15. In the figure, LF is the left front leg of the robot, RF is the right front leg, LR is the left hind leg and RR is the right hind leg. During one movement cycle of the robot, legs and body contact the ground in turn to complete the robot’s movement [26].

Because the robot’s leg drive adopts telescopic electric cylinder, when the supporting leg electric cylinder state is retracted, then the belly of the desilting robot is in contact with the ground, and the body mass of the imitation turtle robot is supported by the body belly. At this time, the robot can save a lot of energy consumption by moving forward. Figure 16 shows a progressive gait cycle of the robot. This gait cycle consists of four movement steps. This design of crawling gait following the design of a turtle belly can improve the stability of the robot, because the projection of the center of gravity of the robot body is always located in the area included by the end line of the robot foot. The robot’s progress is accomplished by changing the length of its swinging legs, which are powered by four legs at the same time and are sufficient to carry heavy equipment.

The robot moves forward in the direction of contact with the ground alternately by its four legs and the body’s abdomen. Each moving gait can be decomposed into the following four movement steps periodically. In step (a), the body of the desiccating robot lands on the belly, its four supporting legs are in the retracted state and separated from the ground, the electric cylinders of the swinging legs on both sides of the front legs are gradually shortened, and the electric cylinders of the swinging legs on both sides of the hind legs are gradually extended, and the motion state of the robot is gradually transformed into step (b). In step (b), the electric cylinders of the four supporting legs of the robot gradually extend. The electric cylinders on both sides of the swinging legs of the front legs gradually extend, and the electric cylinders on both sides of the swinging legs of the hind legs gradually shorten. At this time, the abdomen of the body begins to separate from the ground, and the robot’s four supporting legs support the robot body. In step (c), the electric cylinder of the front swinging leg of the robot is extended, the electric cylinder of the rear swinging leg is shortened, and the robot moves forward. In step (d), the robot needs to change the length of the electric cylinder to bring the body back to the initial state in step (a).

### 4.2. Foot Trajectory Planning

Foot trajectory function is the key factor affecting the robot performance. Planning the proper foot trajectory curve is not only beneficial to improve the robot’s motion characteristics, but also can reduce the robot’s energy consumption. Foot trajectory planning is the process of applying mathematical functions to foot trajectory coordinates. In the process of foot trajectory planning, the coordinate corresponding to the Angle of each joint of the legs when the robot moves is usually solved first, then the length transformation function of each electric cylinder is obtained by inverse kinematics solution, and then the control function of the length change of electric cylinder is written. The solving process of foot trajectory planning is shown in Figure 17.

Using the starting position of the robot’s foot as the origin to establish a coordinate system, the constraint conditions that the robot should satisfy in the x-axis direction during single-leg swing can be expressed using Equation (22). Substituting the constraint conditions into the cubic polynomial *x* = *at*^3^ + *bt*^2^ + *ct* +*d*, the displacement function of the swinging leg in the *x*-axis direction can be obtained as shown in Equation (23).
(22)$$\{\begin{array}{c} x{|}_{t=0}=0\;x{|}_{t=T_{w}/2}=\frac{L}{2}\;x{|}_{t=T_{w}}=L\\ \overset{•}{x}{|}_{t=0}=\overset{•}{x}{|}_{t=T_{w}}={\overset{••}{x|}}_{t=0}={\overset{••}{x|}}_{t=T_{w}}=0 \end{array}$$
(23)$$x(t)=-\frac{2L}{{T_{w}}^{3}}t^{3}+\frac{3L}{{T_{w}}^{2}}t^{2}\;\;\left(0\leq t\leq T_{w}\right)$$
where: the period of oscillating action is *T_w_*, and the unit is s; the time is *t* and the unit is s; the step distance in the *x*-axis direction is *L*, in m; the first derivative of the foot’s *x*-axis coordinates with respect to time is velocity $\overset{•}{x}$, in m/s; the second derivative of the displacement with respect to time is the acceleration $\overset{••}{x}$, and the units are m/s^2^.

During the foot tip swinging process, the foot tip is first lifted vertically to the highest point and then dropped. The constraint condition of the foot tip in the z-axis direction is given by Equation (23). To meet the characteristics of smooth swinging speed and low impact, a piecewise method is used to substitute the constraint condition into the general form of a cubic polynomial to obtain the displacement function of the foot tip in the z-axis as shown in Equation (24):(24)$$\{\begin{array}{lllll} z{|}_{t=0}=0 & z{|}_{t=T_{w}/4}=\frac{H}{2} & z{|}_{t=T_{w}/2}=H & z{|}_{t=3T_{w}/4}=\frac{H}{2} & z{|}_{t=T_{w}}=0\\ \overset{•}{z}{|}_{t=0}=0 & & \overset{•}{z}{|}_{t=T_{w}/4}=V_{\max} & & \overset{•}{z}{|}_{t=T_{w}}=0 \end{array}$$
(25)$$z(t)=\left\{ \begin{array}{ll} -\frac{16H}{{T_{w}}^{3}}t^{3}+\frac{12H}{{T_{w}}^{2}}t^{2} & \left(0\leq t<T_{w}/2\right)\\ H+\frac{16H}{{T_{w}}^{3}}{\left(t-\frac{T_{w}}{2}\right)}^{3}-\frac{12H}{{T_{w}}^{2}}{\left(t-\frac{T_{w}}{2}\right)}^{2} & \left(T_{w}/2\leq t<T_{w}\right) \end{array}\right.$$
where: the period of oscillating action is *T_w_*, and the unit is s; The time is *t* and the unit is s; The step height in the direction of the *z*-axis is *H*, in m; The first derivative of the *z*-axis of the foot with respect to time is velocity $\overset{•}{z}$, in m/s; The second derivative of displacement with respect to time is $\overset{••}{z}$, in m/s_2_.

The constraint conditions of the compound cycloid function in the *x*-axis direction are consistent with that of the cubic polynomial curve. The position function can be obtained by substituting the constraint conditions into the acceleration function $\overset{••}{x}=A_{x}\sin\frac{2\pi t}{T_{0}}$ in the *x*-axis direction and integrating continuously, as shown in Equation (24);
(26)$$x(t)=L\left(\frac{t}{T_{w}}-\frac{1}{2\pi }\sin\left(2\pi \frac{t}{T_{w}}\right)\right)\;\;\left(0\leq t\leq T_{w}\right)$$
where: the swing action period is *T_w_* in s; the time is *t* in s; and the step distance in the *x*-axis direction is *L* in m.

After the *x*-axis position function has been obtained, the *z*-axis function expression needs to be determined in order to achieve a complete foot-end planning trajectory curve. Substitute the constraints into the *z*-axis direction acceleration function $\overset{••}{z}=A_{z}\sin\frac{4\pi t}{T_{0}}$ and integrate continuously to obtain the position function as in Equation (25);
(27)$$z(t)=\{\begin{array}{cc} 2H\left(\frac{t}{T_{w}}-\frac{1}{4\pi }\sin\left(4\pi \frac{t}{T_{w}}\right)\right) & \left(0\leq t<\frac{T_{w}}{2}\right)\\ 2H\left(\frac{T_{w}-t}{T_{w}}+\frac{1}{4\pi }\sin\left(4\pi \frac{t}{T_{w}}\right)\right) & \left(\frac{T_{w}}{2}\leq t<T_{w}\right) \end{array}$$
where: the swing action period is *T_w_* in s; the time is *t* in s; and the step distance in the *z*-axis direction is *H* in m.

To verify the correctness of the foot end planning curve, set the swing period *T_w_* =1 s, step length *L =* 1 m, step height *H* = 0.3 m, and substitute the parameters into the cubic polynomial curve equation *x*-axis displacement Formula (23) and the compound cycloid curve equation *x*-axis displacement Formula (26) to obtain the change in the foot end displacement *x*-axis direction component as shown in Figure 18a. The displacement change in the vertical direction is solved in the same way as in the horizontal direction. Then, substitute the parameters into the cubic polynomial curve equation *z*-axis displacement formula compound cycloid curve Formula (25) and the compound cycloid curve equation *z*-axis displacement Formula (27), to obtain the change in the foot end displacement *z*-axis direction component as shown in Figure 18b.

In Figure 18a, it can be seen that both foot end planning curves meet the previously proposed constraints of position, velocity, and acceleration. When the foot end uses a compound cycloid trajectory, the velocity of the foot end during the initial stage and the end of the swing is slower than when using a cubic polynomial. As time increases, its velocity and acceleration also increase. When the swing time *t* = 0.5 s is reached, the foot end has the same displacement component in the *x*-axis direction for both trajectory planning curves. Figure 18b plots the change trend of the displacement *z*-axis component of the two foot end planning curves. The velocity of both foot end trajectory planning curves shows an acceleration to a maximum and then a deceleration. When the swing time *t* = 0.5 s, the foot end reaches its highest point in the *z*-axis direction. After *t* = 0.5 s, the foot end begins to descend, with a velocity change trend similar to that of the lifting phase. When *t* = 1 s, the step action is completed.

In order to more intuitively observe the trajectory of the swing process of the foot, the *x*-axis displacement of the foot is taken as the horizontal axis and *z*-axis displacement of the foot is taken as the vertical axis to draw the foot trajectory curve, as shown in Figure 19. Figure 19 shows that when the foot trajectory planning is carried out by using the compound cycloidal curve and cubic polynomial curve, the foot trajectory paths planned by the two curves are basically consistent at the initial lifting stage of the swinging foot, and the swing range of the compound cycloidal trajectory is slightly larger than that of the cubic polynomial curve after the foot is lifted. Generally speaking, there is little difference between the two. Both of them can enable the robot foot to have the ability to cross certain obstacles [27].

### 4.3. Robot Energy Consumption Model and Energy Consumption Evaluation Index

The total energy consumption of the robot mechanical system within a gait cycle is the integral of the instantaneous power of the electric cylinder against time at each moment. Given the force of the electric cylinder and the speed of the electric cylinder actuator, assuming, that the driving force does not do negative work, the total energy consumption of the bionic turtle belly crawling mechanism can be expressed in Equation (26). Obviously, when different foot trajectory functions or different gait parameters are selected, the force and speed of each electric cylinder actuator are different, so the energy consumed by the corresponding system is also different.
(28)$$E=\sum_{i=1}^{4}\sum_{j=1}^{2}\int_{0}^{Tw}\left|F_{jw}V_{ijw}(t)\right|dt+\sum_{i=1}^{4}\sum_{j=1}^{2}\int_{Tw}^{T}\left|F_{js}V_{ijs}(t)\right|dt$$
where: *F_jw_* is the force on the *j*-th electric cylinder in the oscillating phase, N; *V_ijw_* is the *j*-th electric velocity of the *i*-th leg in the swing phase, m/s; *F_js_* is the force on the *j*-th electric cylinder in the support phase, N; *V_ijs_* is the *j*th electric velocity of the *i*-th leg in the oscillating phase, m/s.

The energy consumption evaluation of the robot is mainly evaluated from the aspects of energy consumption ratio, average power ratio, power density loss and so on. The above three evaluation indexes are described. 

Energy consumption ratio of robot: The energy consumed by moving a unit weight object per unit distance is the moving energy consumption rate, also known as the energy consumption ratio. The energy consumption ratio pays more attention to the analysis of the entire energy consumption results. The smaller the energy consumption ratio value, the better the robot energy utilization [28,29,30,31]. The energy consumption ratio can be expressed in Equation (27):(29)$$\varepsilon =\frac{E}{mgs}$$
where: the energy consumed by the robot during its movement is *E* in J; the body mass of the robot is *m*_0_ in kg; the acceleration of gravity is *g* in m/s^2^; and the distance travelled is *s* in m.

### 4.4. Robot Adams Simulation Experiment and Result Analysis

Figure 20 shows the simulation sequence of one abdominal crawling gait cycle of the tortoise-like robot:

At T = 0~1 s, the robot is located in the initial state, and the robot support legs are in contact with the ground, carrying the weight of the body.

At T = 1~2 s, the front swing leg is extended, the rear swing leg is shortened, and the body moves forward.

At T = 2~3 s, both the front and rear support legs are retracted and the abdomen of the robot is supported on the ground.

At T = 3~4 s, after completing the intended foot-end trajectory planning curve, the support leg makes contact with the ground.

The main factors that have a significant impact on the energy consumption of a robot are the structure of the robot, the foot-end trajectory function and the gait parameters. In this paper, the structure and size of the robot have been determined, so the gait parameters were explored and two gait parameters, step length and step height, were selected to optimise the robot to reduce its energy consumption.

To control the variables robot simulation period and step height should be consistent, step height is set to 0.2 m constant, step length selection range refer to foot end kinematics analysis. It was set to vary from 0.2 m to 0.8 m, and the simulation gait energy consumption experiment was set every 0.1 m. The Adams simulation results were imported into Matlab for processing and the resulting fitted curve of step length to robot energy consumption ratio is shown in Figure 21.

The energy consumption ratio of the robot can reflect the energy utilisation of the robot. As can be seen from Figure 21, the energy consumption ratio of the robot gradually decreases as the step length increases, and the ratio decreases faster when the step length is 0.2~0.4 m. When the step length range is 0.5~0.8 m, the energy consumption ratio of the robot gradually tends to be stable as the step length increases, and the energy consumption of the compound pendulum foot-end trajectory is greater than that of the cubic polynomial trajectory.

To control the variable robot simulation period and step length should be consistent, the step length was set to 0.5 m constant, and the step height selection range was also referred to the robot kinematics analysis. The step height was set to vary from 0.1 m to 0.3 m, with each 0.05 m increase set for one simulation experiment, and the obtained step height and robot energy consumption curves are shown in Figure 22.

As depicted in Figure 22, when the step length is maintained at 0.5 m, the energy consumption of the walking robot increases with an elevated step height ranging from 0.1–0.3 m. Consequently, the energy consumption ratio gradually rises, indicating an augmented energy expenditure per unit mass during locomotion. Simultaneously, a decrease in the disparity of energy consumption ratios suggests a convergence between gaits as step height decreases. Notably, adopting a compound cycloidal foot trajectory results in higher energy consumption compared to a cubic polynomial foot trajectory due to its superior obstacle-crossing capability during single-leg swing phase and subsequent increased energetic demands.

## 5. Robot Experiment

### 5.1. Robot Step Test

The actual size of the robot required for the project is large, and taking into account the processing cost and cycle time of the experimental setup, as well as the purpose of the experiment, simulations were carried out based on similar theory, and the experimental prototype device was manufactured using a 1/5 scale scaling [32]. The main purpose of the experimental session was to explore the characteristics of the robot gait in relation to energy consumption. The dredging part was therefore simplified to reduce processing costs, and the dredging mechanism and auxiliary pressurisation device were omitted from the experimental prototype. The overall layout of the experimental isoscale prototype is consistent with the engineering design of the turtle abdominal crawling mechanism. The prototype experimental setup is shown in Figure 23, in order to reduce the weight of the robot, the main body of the prototype was made of aluminium alloy and the key parts were made of Q235. The components used in the experimental prototype are the same as the control principles required for the project. In summary, the experimental prototype device has been designed to verify the theoretical analysis. The experimental prototype will be used as the basis for subsequent experiments on the effect of the robot foot trajectory function, step length and step height parameters on the energy consumption of the robot.

As the control function obtained from the robot foot-end planning is more complex, in order to simplify the writing of the control program, the function calculated by the theory needs to be simplified, and the function written after the simplified fitting is downloaded to the robot control board, and the control board can calculate the number of pulses to be output according to the function, and the pulses are transmitted to the robot drive board to complete the motor drive. The experimental prototype was photographed and recorded as shown in Figure 24 The solid line in Figure 24 is the ground stationary reference datum line and the dashed line is the center line of the half of the body position.

### 5.2. Phase Synchronization Step Size Experiment with High Parameters

The step length of the prototype was incremented by 25 mm at a time, gradually increasing from 50 mm to 200 mm, the step height parameter was chosen to be 40 mm constant and the gait traveled for 10 cycles. The energy consumed by the experimental prototype with a compound cycloidal trajectory at the foot end and the calculated energy consumption ratio are shown in Table 2.

The second group explored the effect of step length parameters on the energy consumption of the robot The experimental procedure was the same, with step length increments of 25 mm at a time, gradually increasing from 50 mm to 200 mm, step height parameters chosen to be constant at 40 mm and gait walking for 10 cycles. The energy consumed by the experimental prototype with a cubic polynomial trajectory at the foot end and the calculated energy consumption ratio are filled in as shown in Table 3.

Table 2 and Table 3 show that in terms of energy consumption: both foot-end trajectory functions show an increase and then a decrease in energy consumption as the step length increases. When using a compound cycloidal foot-end trajectory the robot consumes more energy than when using a cubic polynomial foot-end trajectory. The experimental data for robot step length and step height obtained by plotting the composite cycloidal foot-end trajectory function against the cubic polynomial foot-end trajectory function is shown in Figure 25.

The energy consumption ratio decreases rapidly with an increase in the robot’s step length during the initial stage, as depicted in Figure 25. The curve decline speed is initially rapid and then slows down, resulting in a diminishing difference in energy consumption ratio. Based on the trend observed in the figure, when the step length reaches a sufficient magnitude, the difference in energy consumption ratio can be considered negligible, leading to overlapping curves. Notably, planning the robot foot as a compound cycloidal trajectory function results in higher energy consumption ratio compared to planning it as a cubic polynomial trajectory function. Furthermore, with an increasing step length, the robot’s energy consumption ratio tends to stabilize.

### 5.3. Prototype Step Height Experiment with the Same Step Parameters

The step height of the prototype was incremented by 10 mm at a time, gradually increasing from 20 mm to 60 mm, the step length parameter was chosen to be 100 mm constant and the gait was repeated for 10 cycles of the experiment. The energy consumed by the experimental prototype with a compound cycloidal trajectory at the foot end and the calculated energy consumption ratio are shown in Table 4.

The second group investigated the effect of step height parameters on the robot’s energy consumption similar to the first group, again incrementing the step height by 10 mm at a time, gradually increasing from 20 mm to 60 mm, choosing a constant step length parameter of 100 mm and planning the gait for 10 cycles. A cubic polynomial trajectory was used for foot-end trajectory planning, and the experimental results are shown in Table 5.

From Table 4 and Table 5, it can be seen that in terms of energy consumption, both foot-end trajectory functions show a gradual increase in energy consumption as the step height increases, with the composite cycloidal foot-end trajectory function consuming more energy than the cubic polynomial foot-end trajectory. The experimental data on robot step length and step height obtained by plotting the composite cycloid foot-end trajectory function against the cubic polynomial foot-end trajectory function are shown in Figure 26.

The increase in the robot’s step height, as depicted in Figure 26, leads to a corresponding rise in the robot’s energy consumption ratio. The overall curve exhibits an initial deceleration followed by acceleration. Initially, the energy consumption ratio increases gradually with step height, but it accelerates at a faster rate as the step height further increases. In comparison to the simulation experiment, the compound cycloid still demonstrates higher energy consumption than the cubic polynomial with increasing step height, consistent with simulation results. However, there is minimal variation in the difference of energy consumption ratios within the experimental range of step heights. Towards the end of our data collection period, this difference begins to decrease slightly, aligning closely with our simulation process. This limitation arises due to constraints on mechanical structure height. Tus, only a maximum step height of 60 mm can be achieved for our experimental model prototype.

The energy consumption ratio of the robot is related to the mass of the robot, as can be seen from the energy consumption ratio formula. The energy consumption ratio of the simulated model and the experimental model follows the same trend as the gait parameters, but the energy consumption ratio varies significantly from the simulated results. This is due to the aluminium alloy used for the robot and the simplified structure of the prototype. When the prototype is scaled to a 5:1 ratio, the mass of the prototype is not consistent with the geometric scaling, and the mass is reduced by a factor of about 10. This paper focuses on the trend of the influence of gait parameters on the energy consumption ratio, so the experimental results of the prototype meet the experimental requirements.

## 6. Conclusions

The simulation experiment and the equally scaled prototype experiment both confirmed that, when considering the same gait parameters, the energy consumption of the compound cycloid trajectory was higher than that of the cubic polynomial foot trajectory. For planning the foot trajectory function of the bionic turtle belly crawling desilting robot, a cubic polynomial function can be employed to reduce energy consumption. The analysis of energy characteristics revealed that as step length increased, the energy consumption ratio decreased; conversely, as step height increased, the energy consumption ratio increased. Therefore, in designing gait for the bionic turtle belly crawling robot, it is recommended to opt for longer step lengths and lower step heights to enhance energy utilization while ensuring obstacle crossing performance.

## Figures and Tables

**Figure 1 biomimetics-09-00147-f001:**
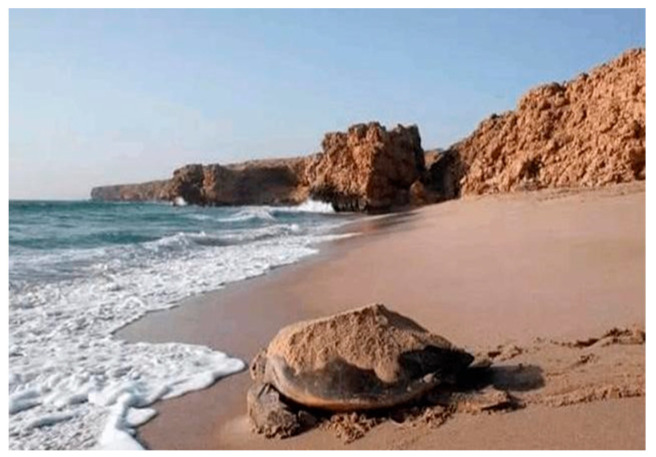
Turtle crawling.

**Figure 2 biomimetics-09-00147-f002:**
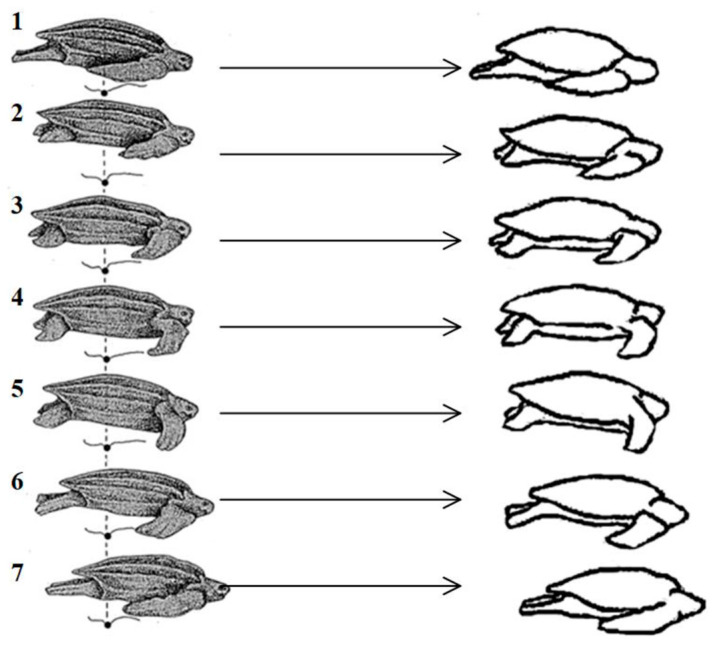
Order of turtle crawling gait movement.

**Figure 3 biomimetics-09-00147-f003:**
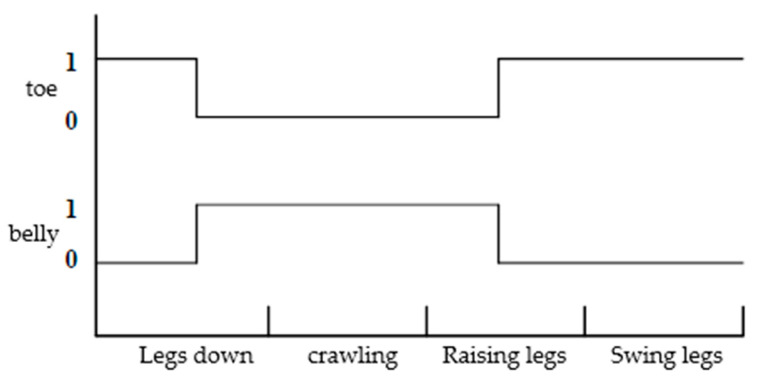
Phase diagram of the turtle’s abdominal crawling gait.

**Figure 4 biomimetics-09-00147-f004:**
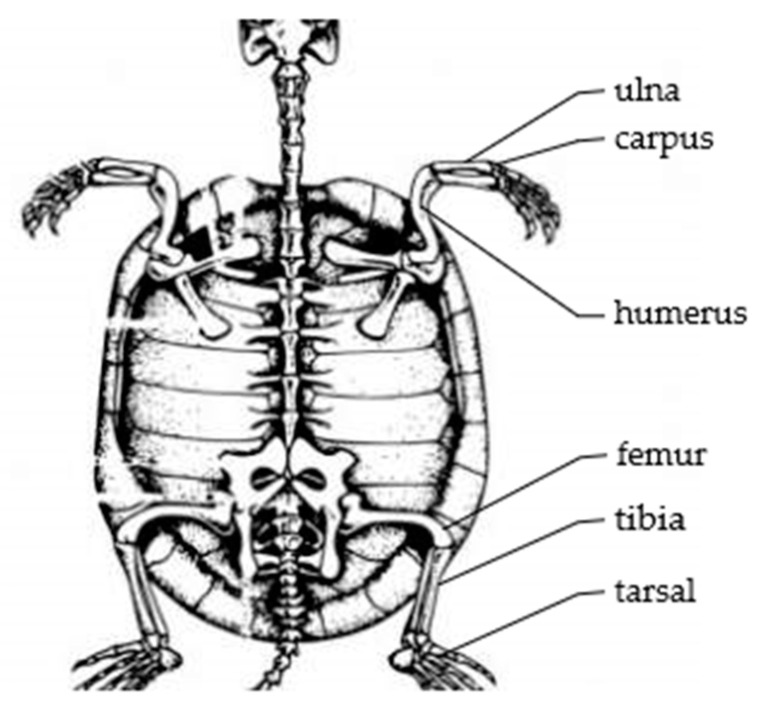
Structural diagram of part of the turtle’s skeleton.

**Figure 5 biomimetics-09-00147-f005:**
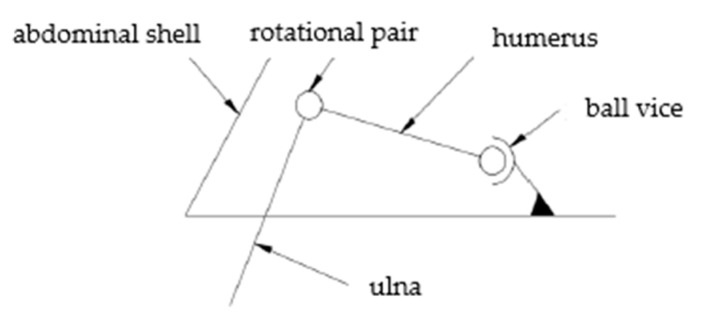
Structural relationship of single leg joint of sea turtle.

**Figure 6 biomimetics-09-00147-f006:**
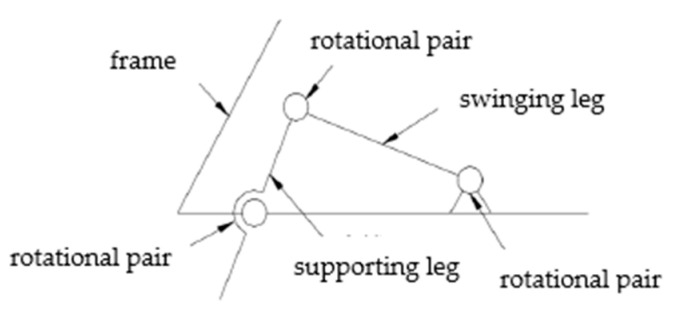
Diagram of the structure of the legs of the crawling robot for dredging silt.

**Figure 7 biomimetics-09-00147-f007:**
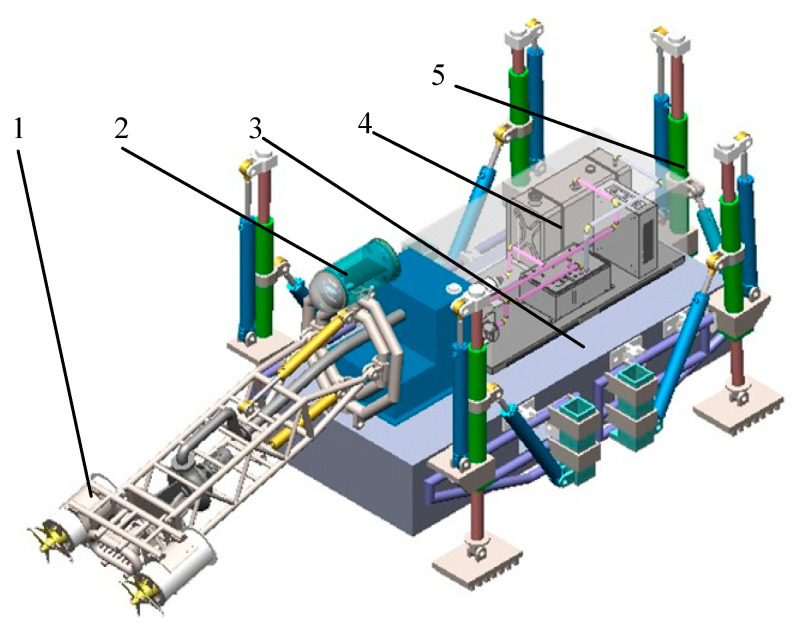
Three-dimensional diagram of the dredging robot. 1—Dredging executive device 2—underwater electronics cabin 3—turtle imitation robot body 4—auxiliary booster device 5—rear steering leg.

**Figure 8 biomimetics-09-00147-f008:**
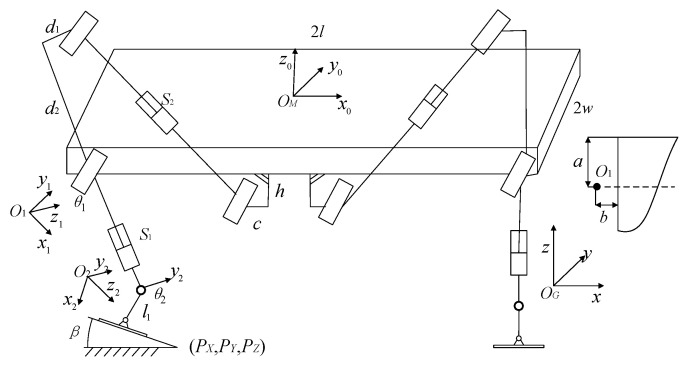
Simplified kinematics model of the robot.

**Figure 9 biomimetics-09-00147-f009:**
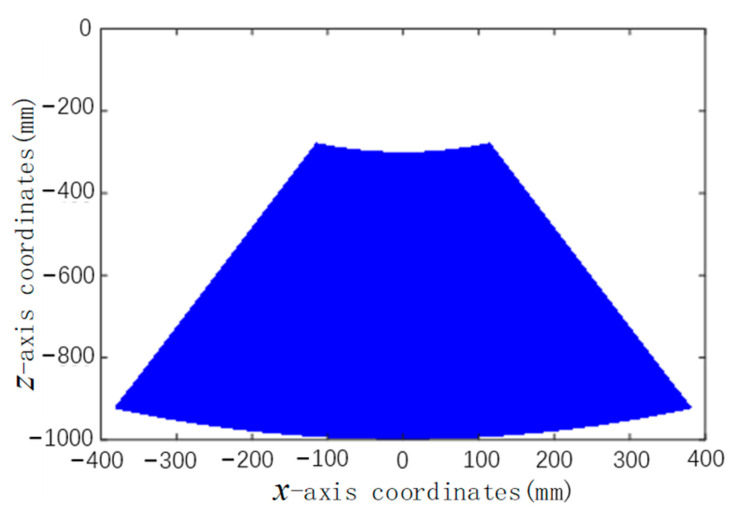
Swing space diagram of the foot of the supporting leg.

**Figure 10 biomimetics-09-00147-f010:**
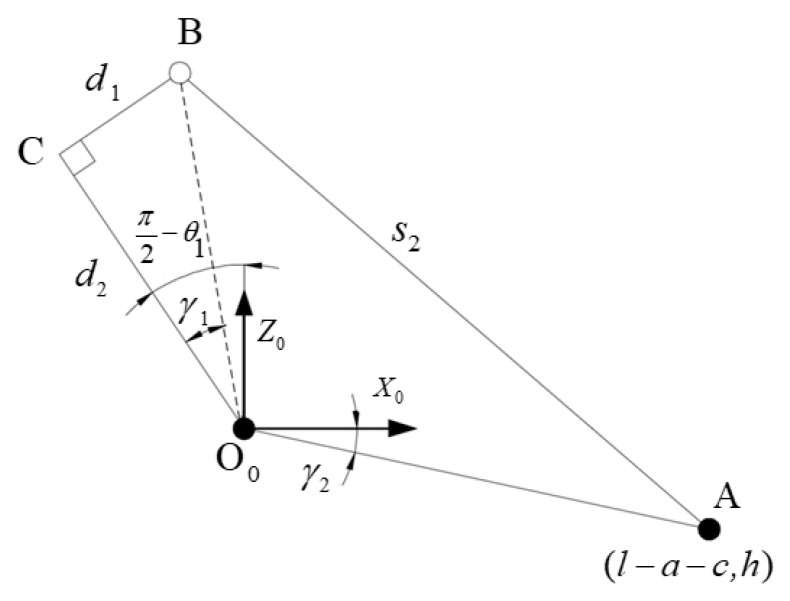
Inverse kinematics geometric analysis model of robot legs.

**Figure 11 biomimetics-09-00147-f011:**
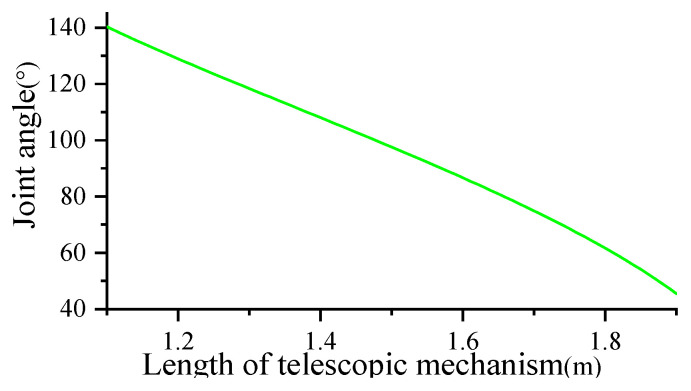
Relation curve between swing of supporting leg *θ*_1_ and length of swinging leg *s*_2_.

**Figure 12 biomimetics-09-00147-f012:**
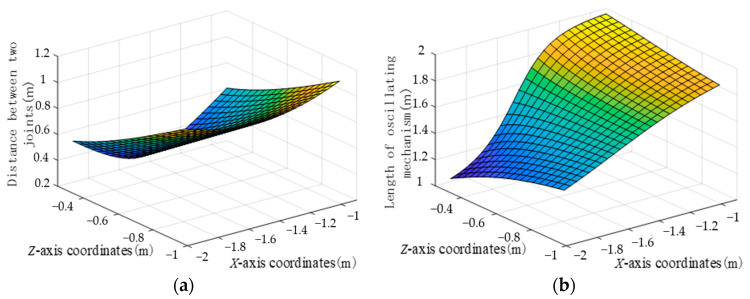
Relation between foot coordinates and robot leg length: (**a**) Foot end coordinates in relation to support leg *s*_1_; (**b**) Foot end coordinates in relation to swing leg *s*_2_.

**Figure 13 biomimetics-09-00147-f013:**
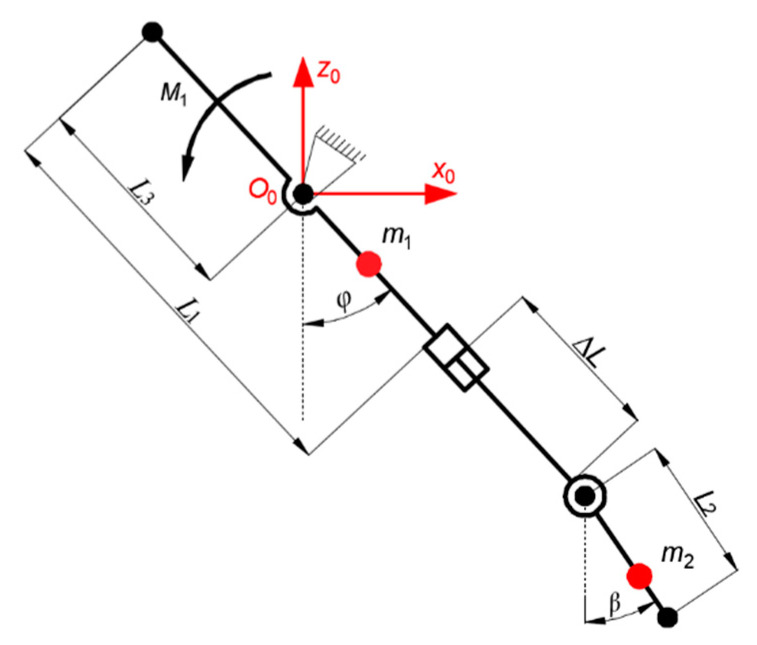
Single leg dynamics model.

**Figure 14 biomimetics-09-00147-f014:**
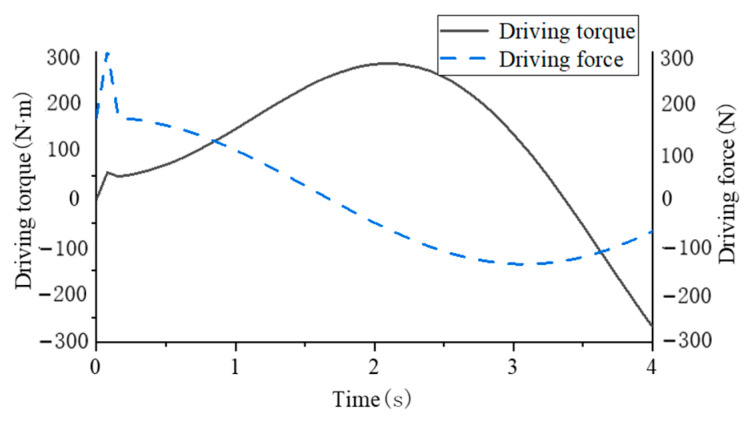
Relation curve between swing of supporting leg *θ*_1_ and length of swinging leg *s*_2_.

**Figure 15 biomimetics-09-00147-f015:**
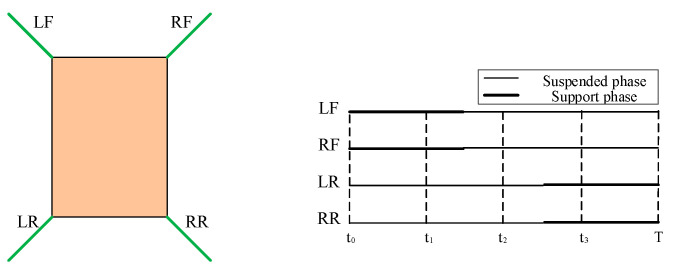
Phase diagram of crawling gait of imitation tortoise on abdomen.

**Figure 16 biomimetics-09-00147-f016:**
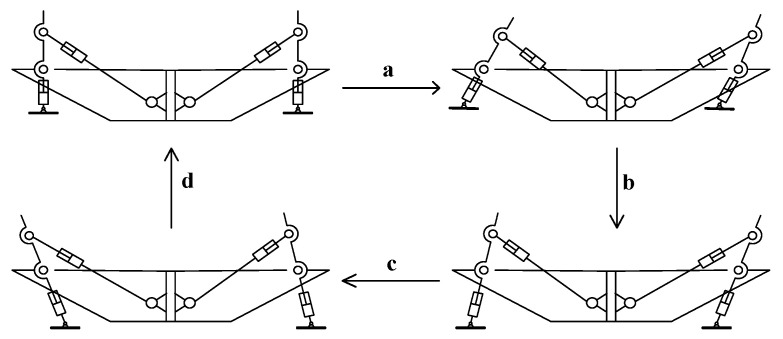
Crawling gait of imitation turtle.

**Figure 17 biomimetics-09-00147-f017:**
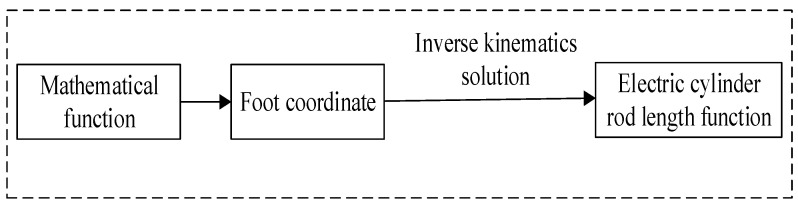
Solving process of foot trajectory.

**Figure 18 biomimetics-09-00147-f018:**
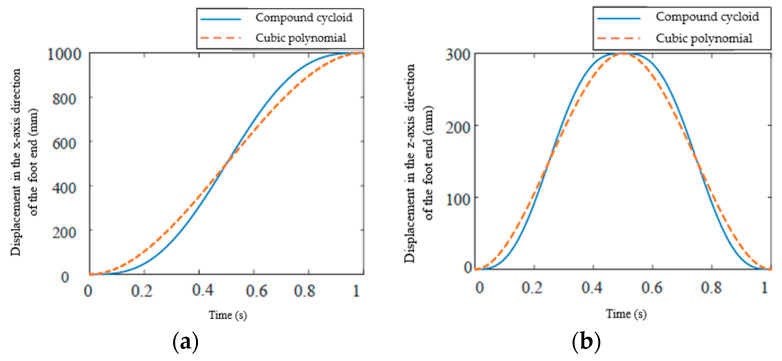
Foot displacement and time graph: (**a**) Displacement in the *x*-axis direction of the foot end; (**b**) Displacement in the *z*-axis direction of the foot end.

**Figure 19 biomimetics-09-00147-f019:**
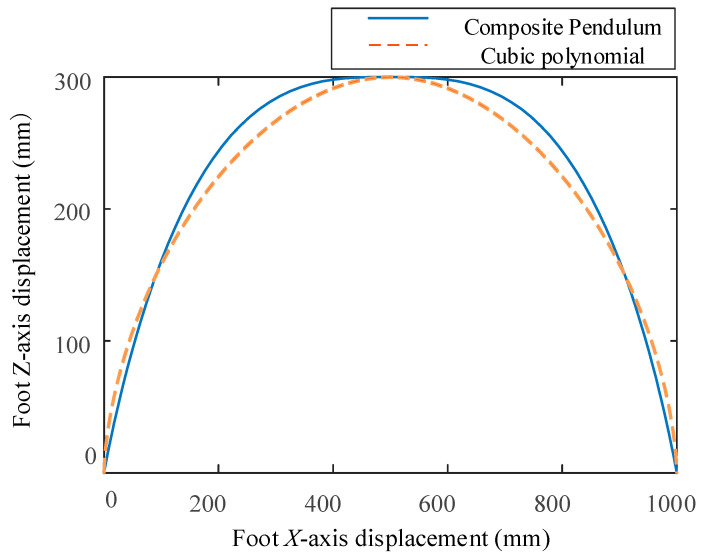
Curve diagram of foot trajectory.

**Figure 20 biomimetics-09-00147-f020:**
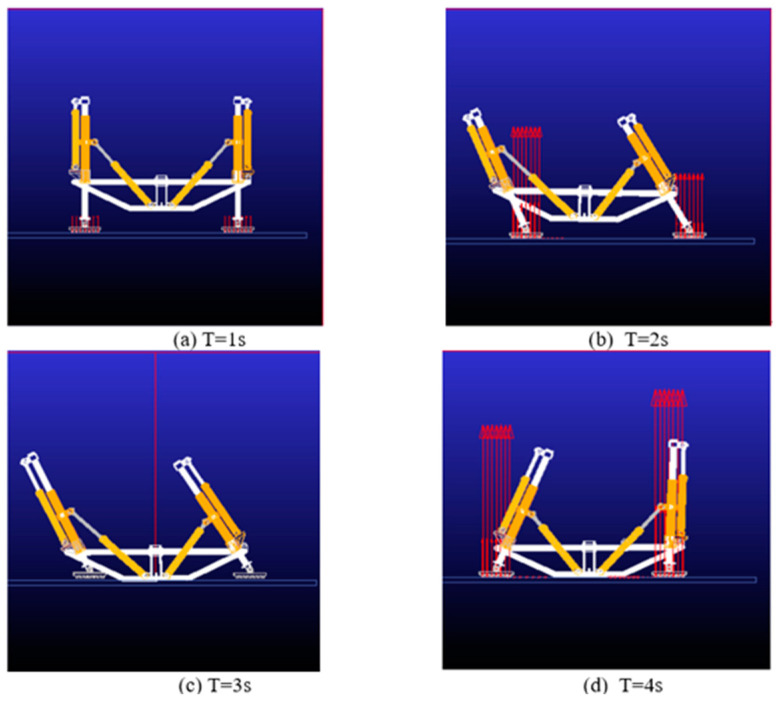
Abdominal crawling gait.

**Figure 21 biomimetics-09-00147-f021:**
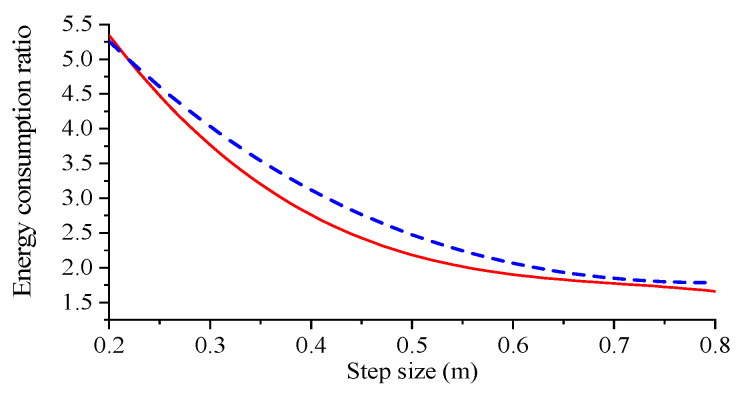
Robot step size to energy consumption ratio.

**Figure 22 biomimetics-09-00147-f022:**
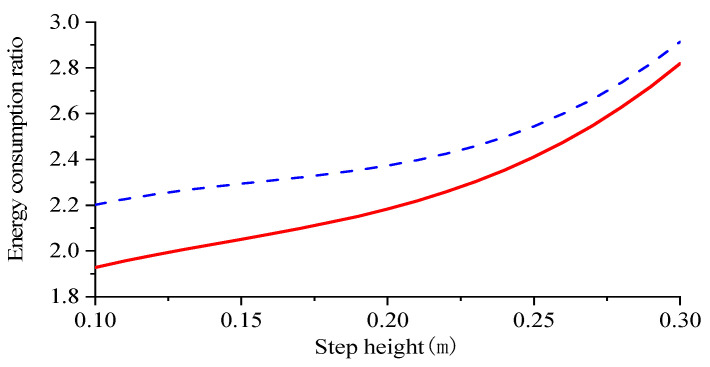
Robot step height to energy consumption ratio.

**Figure 23 biomimetics-09-00147-f023:**
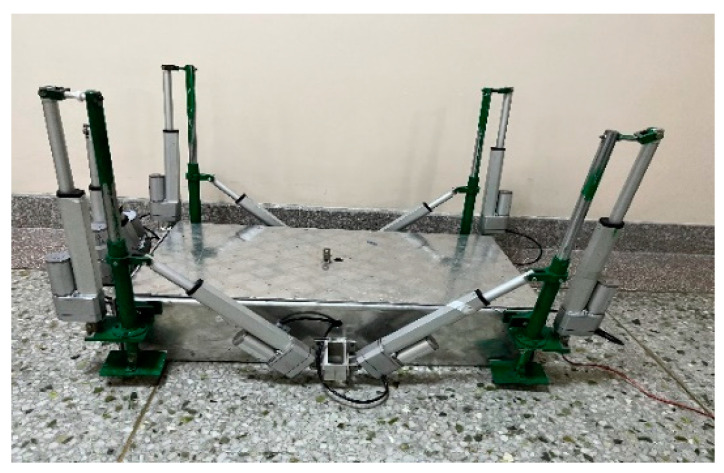
Prototype robot.

**Figure 24 biomimetics-09-00147-f024:**
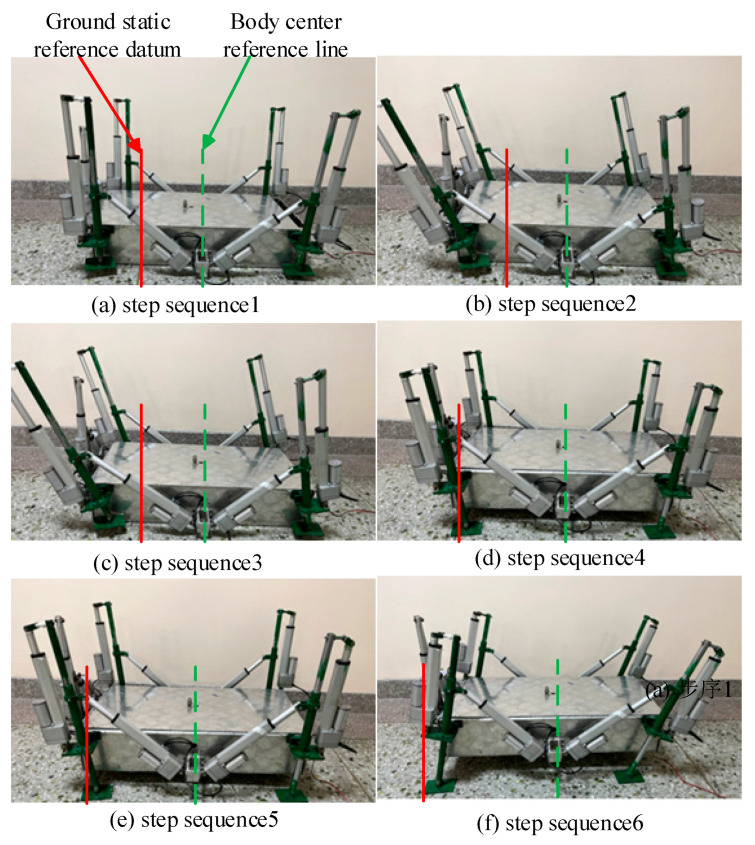
Experimental gait diagram of the prototype’s abdominal crawl.

**Figure 25 biomimetics-09-00147-f025:**
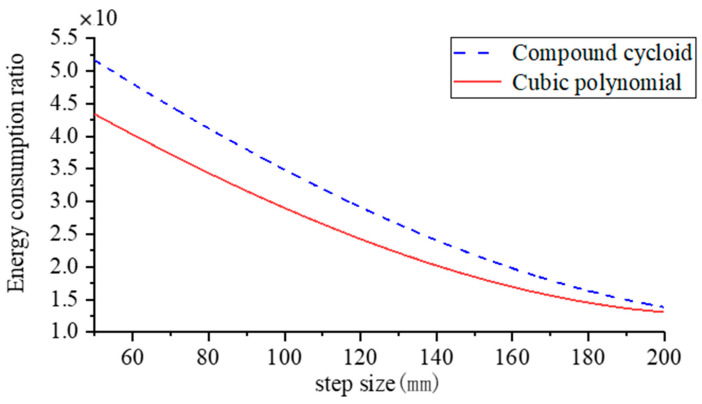
Prototype step size to energy consumption ratio.

**Figure 26 biomimetics-09-00147-f026:**
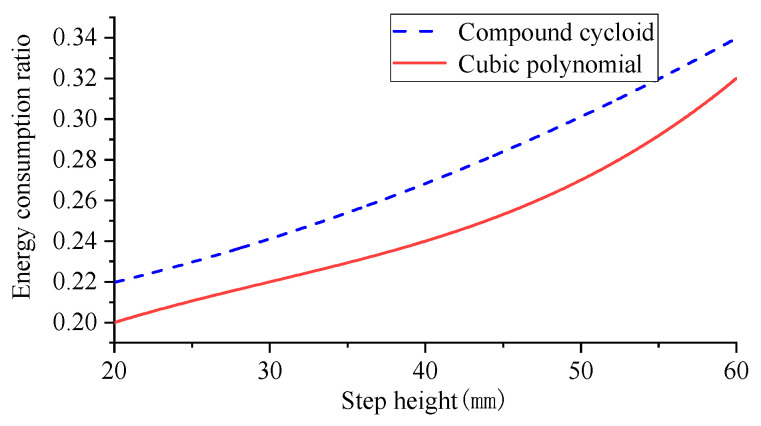
Prototype step height to energy ratio.

**Table 1 biomimetics-09-00147-t001:** Variable parameters of kinematic model of turtle crawling mechanism.

Parameter Symbol	Parameter Meaning	Numerical Value
l1	The distance between the center of the upper plantar face of the robot and the ankle joint	60 mm
d1	Vertical distance between the articulation point of the upper side of the supporting leg and the hip joint	220 mm
d2	Distance between the upper lateral articulation point and the hip joint parallel to the direction of the supporting leg	510 mm
h	Distance between the hinge point of the swing leg and the plane of the body	580 mm
c	The distance between the swing leg hinge point and the central cross section of the body	100 mm
a	Distance from hip joint to front end of body	150 mm
b	Distance from hip joint to lateral surface of body	165 mm
l	1/2 the length of the robot	1600 mm
w	1/2 the width of the robot	700 mm
θ_1_	Swing Angle of support leg	−0.75π~−0.25π

**Table 2 biomimetics-09-00147-t002:** Experiments with foot-end trajectory steps of composite pendulums.

Step Parameter (mm)	Consume Energy (J)	Energy Consumption Ratio
50	10,458	52.29
75	12,814	42.71
100	14,046	35.12
125	14,622	29.24
150	12,749	21.25
175	12,131	17.33
200	11,585	14.48

**Table 3 biomimetics-09-00147-t003:** Trivial polynomial foot-end trajectory step length experiments.

Step Parameter (mm)	Consume Energy (J)	Energy Consumption Ratio
50	8641	43.21
75	11,386	37.95
100	11,331	28.33
125	11,786	23.57
150	11,553	19.26
175	10,784	15.41
200	10,629	13.29

**Table 4 biomimetics-09-00147-t004:** Compound pendulum foot-end trajectory step height experiments.

Step Height Parameter (mm)	Consume Energy (J)	Energy Consumption Ratio
20	8846	22.12
30	9732	24.33
40	11,041	27.60
50	12,138	30.34
60	13,763	34.41

**Table 5 biomimetics-09-00147-t005:** Trivial polynomial foot-end trajectory step height experiment.

Step Height Parameter (mm)	Consume Energy (J)	Energy Consumption Ratio
20	8207	20.52
30	9113	22.78
40	9681	24.20
50	10,957	27.39
60	13,104	32.76

## Data Availability

The datasets used or analysed during the current study are available from the corresponding author on reasonable request.

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
