# Peer review of "Kinematics Analysis and Gait Study of Bionic Turtle Crawling Mechanism"

_biomimetics, 2024, doi:10.3390/biomimetics9030147_

Round 1
Reviewer 1 Report
Comments and Suggestions for Authors
Dear authors,
The topic addressed by the article is interesting, but many unjustified methodological choices make it necessary to improve the article for publication. Authors might consider the following questions, remarks and comments:
- - The objective is not clearly presented. Is it the presentation of a bioinspired turtle robot? A morphologically inspired locomotion with minimized energy consumption? The authors do not propose any hypotheses for this interesting work. There's no link with animal biomimetism. The theoretical considerations put forward are purely robotic and based on what already exists in terms of underwater robots, mechanical structures and control laws. The introduction should be rewritten with this in mind, and references to turtle morphology, locomotion and energy consumption should be added to justify a bioinspired robot.
- - The presentation of morphological and anatomical data on a turtle could have been used to justify the biomimetics of the proposed dredging robot. Why not justify the robot's mechanical structure on biological and morphological grounds? How original is the mechanical structure of the proposed robot? Why is the choice of the number of degrees of freedom and the type of mechanical linkage not justified? These points seem to me essential to improve the originality and biomimetic relevance of the article.
- - The presentation of the morphological properties of a turtle should be added to justify the choice of trajectories used for the robot's locomotion. Ideally, data from real turtle trajectories should be available for comparison with those of the robot? Changes need to be made to all the paragraphs, as this point is central and essential to the claim of a robotic structure bioinspired, otherwise it's classic robotics.
- - Also missing are numerous references to the bioinspired trajectory synthesis used for quadruped robots.
- - The energetic analysis of the robot is interesting, but comes on top of a very robotic description. Is the aim of the work to propose an energetic index? Or to present a turtle robot?
Author Response
1.Thank you for your suggestion. This article is based on the motion gait of sea turtles and designs a biomimetic sea turtle robot for clearing tunnels. However, the focus of the work is to find the optimal and most energy-efficient robot gait through simulation calculations and comparison of various gaits. After reading your suggestion, I will also add more evidence related to biomimetics and provide more information on our work.
2.Regarding your second suggestion, we will provide work content in the article to demonstrate the biomimetic nature of robots and demonstrate the rationality of mechanical structures from both biological and morphological perspectives. Regarding novelty, previous biomimetic turtle robots used rotational power to drive joints. We designed a push-pull form, which brings new problems in simulation and control. We have also solved these problems, but the methods to solve these problems are not the focus of this article. In addition, in order to apply biomimetic foot robots to tunnel cleaning, we have also designed many new structures that can match the robot. The structure and working mode of the entire robot are also innovative. We will also explain in the article the selected degrees of freedom for merging connecting rods.
3.Thank you very much for your suggestion. We will add content on the morphological characteristics and gait analysis of sea turtles in the article to demonstrate its biological inspiration.
4.Thank you for your suggestion. We will add relevant references.
5.The energy analysis of the article focuses on the gait of each leg of the robot, not the entire robot. The main purpose of this work is to compare and analyze the energy consumption of each gait, identify the gait with less energy consumption, and complete the research content。
Reviewer 2 Report
Comments and Suggestions for Authors
This paper deals with the kinematics analysis and gait study of bionic turtle crawling mechanism and gait, by designing a crawling robot.
I think that this paper should be revised carefully. I suggest the following comments:
1) The authors should polish the paper suitably. The whole paper should be reviewed carefully, to correct all the typing errors. For example in the Abstract, we cannot say “this paper designs ..”
2) The introduction should be further improved and structured. More discussion should be added with more works from the literature.
3) Also, for the introduction, you need to include and emphasize the motivation, the main difficulties, the main work, and the improvements compared with previous related works.
4) The novelty of the proposed method should be highlighted carefully.
5) More figures should be added to demonstrate further the effectiveness of the design method of the turtle crawling robot.
6) In page 19, you have Figure 1.??? Please check the numbers.
7) It seems that there is a divergence between compound cycloid and cubic polynomial in this previous figure and also in Figure 19. Explain why there is a difference.
8) you need to add more details on how you can solve or compute Eq (26)
9) You should explain how you have fixed the trajectories in Eq (22)-(25). The authors should illustrate how to choose these parameters in the different expressions of trajectories.
10) It is not clear the importance of Figure 12. If possible, eliminate it, or you need to redraw it by adding more details inside it.
11) It seems that the designed turtle crawling robot has a walking mechanism or gait like that of the quadruped robots. More details about this can be added in the paper.
Comments on the Quality of English LanguageEnglish should be improved since there are some weak sentences.
Author Response
1.Thank you for your suggestion. We will read the article again and make corrections to any inappropriate or incorrect parts.
2.We will improve the structure of the article based on your feedback, while adding discussions and references.
3.Thank you for this suggestion. In the introduction, we will emphasize the motivation, main problems, main work, etc., and compare our work with previous work to show the progressiveness of our work.
4.The novelty of this article is mainly reflected in the design of the robot structure, which imitates the gait of sea turtles, allowing the robot to work in water channels filled with mud and debris. Then, the robot's gait is planned, and energy analysis is performed on the gait to obtain the optimal gait. The novelty of the part about post gait planning and energy is mainly reflected in the different target objects. In the past, robotic arms or legged robots often used rotation to drive the relevant joints of the robot, while we all used push-pull method to drive the joints of the robot, which brings different boundary adjustments in the simulation process and adds a certain degree of novelty. We will also reflect this in the article.
5.Thank you for your suggestion. We will provide more explanation on the crawling process of sea turtles and the bionic design process of robots in the article.
6.We apologize for the sixth suggestion and will double check the article to correct any errors.
7.Indeed, as you said, there are differences in the results obtained from the two types of line shapes, but there are also inherent differences between the two. These two graphs are the results of energy analysis for two different gait plans. It can be seen from the graphs that the energy consumption ratio of the robot's foot end planning as a composite cycloid trajectory function is higher than that of using a cubic polynomial foot end trajectory. As the step size continues to increase, the energy consumption ratio of the robot tends to stabilize. This result is also one of the important results of the article's work, so differences are normal.
8.Thank you for your suggestion. Equation 26 is the energy consumption model for robots, which itself is a simple physical formula, that is, energy consumption E=P × t. P is the power of the electric cylinder, t is the working time, and P=FV is also a commonly used formula for power. Equation 26 is the integration of each electric cylinder over time, and the result obtained by adding them can be directly written. However, for calculating it, specific data and relevant mathematical tools are required. Therefore, Equation 26 is only the energy consumption model of the robot proposed in the article, which can help readers better understand the energy consumption methods and pathways of the robot's working process.
9.Thank you very much for your suggestion. These methods are ultimately obtained through various boundary conditions, and we will provide specific explanations in the text.
10.Thank you for your suggestion. The image is intended to provide a macro and intuitive explanation of the robot's foot planning approach. After discussion, we have decided to keep it and will provide further explanation.
11.Thank you for your suggestion. We will add more reference content for quadruped robots in the article to explain this aspect.
Round 2
Reviewer 1 Report
Comments and Suggestions for Authors
Thank you for taking our questions and recommendations into account.
Sincerely,
Author Response
Thank you for your review. We will revise the article again according to your suggestions.
Reviewer 2 Report
Comments and Suggestions for Authors
Authors have clearly improved the paper. Nevertheless, the discussion on the divergence in (difference between) the curves in Figures 22, 25, and 26 should be further discussed and clearly explained. it is not clear under its actual presentation.
Comments on the Quality of English LanguageAdequate but can be improved for some possible weak sentences.
Author Response
Thank you for your review. We will revise the article again according to your suggestions, adding relevant explanations about Figure 22, 25, 26 in the article, and refining the article.
Round 3
Reviewer 2 Report
Comments and Suggestions for Authors
Paper clearly improved and then it can be accepted